# Application of Logging While Drilling Tool in Formation Boundary Detection and Geo-steering

**DOI:** 10.3390/s19122754

**Published:** 2019-06-19

**Authors:** Gaoyang Zhu, Muzhi Gao, Fanmin Kong, Kang Li

**Affiliations:** The School of Information Science and Engineering, Shandong University, Qingdao 266237, China; gaoyangzhu@mail.sdu.edu.cn (G.Z.); gaomuzhi@whu.edu.cn (M.G.); kongfm@sdu.edu.cn (F.K.)

**Keywords:** Logging while drilling (LWD), geo-steering, continued fraction, boundary detection, inversion modeling, Levenberg-Marquardt algorithm

## Abstract

Logging while drilling (LWD) plays a crucial role in geo-steering, which can determine the formation boundary and resistivity in real time. In this study, an efficient inversion, which can accurately invert formation information in real time on the basis of fast-forward modeling, is presented. In forward modeling, the Gauss–Legendre quadrature combined with the continued fraction method is used to calculate the response of the LWD instrument in a layered formation. In inversion modeling, the Levenberg–Marquardt (LM) algorithm, combined with the line search method of the Armijo criterion, are used to minimize the cost function, and a constraint algorithm is added to ensure the stability of the inversion. A positive and negative sign is added to the distance parameter to determine whether the LWD instrument is located above or below the formation boundary. We have carried out a series of experiments to verify the accuracy of the inversion. The experimental results suggest that the forward algorithm can make the infinite integral of the Bessel function rapidly converge, and accurately obtain the response of the LWD instrument in a layered formation. The inversion can accurately determine the formation resistivity and boundary in real time. This is significant for geological exploration.

## 1. Introduction

With the development of high angle wells and horizontal wells, logging while drilling (LWD) has received widespread attention for geological exploration [1]. The data used in pre-drilling research are uncertain, which will cause the horizontal wells drilled along the design track to not be in the optimal position in the reservoir, thus affecting the drilling effect of the target formation. LWD data can determine the location of the LWD tool and adjust the trajectory of the wellbore in real time [2]. In the process of high angle and horizontal well drilling, real-time adjustment of the well trajectory based on LWD data can help the oil field to improve the return of the drilling investment [3]. The traditional electromagnetic LWD tool using the axial launching and receiving antenna, does not have azimuth detection characteristics. The instrument used in this study is equipped with a transverse antenna based on traditional instruments. These electromagnetic LWD tools are widely used in geo-steering because the measurement results have azimuth characteristics and can better determine the orientation of the formation boundary [4,5]. They provide formation resistivity evaluation and instrument position information [6]. In the actual measurement while drilling, it is inevitable to encounter the influence of formation factors such as the wellbore and the electrical influence of the instrument itself; therefore, symmetry compensation transmitting coils are added.

The inversion needs to repeatedly call forward modeling; efficient forward modeling is critical to the inversion. Forward modeling can determine the response of the LWD tool in a known formation [7]. The induction logging tool theory was originally developed by Doll [8]. It only considers a tool in a vertical borehole. Reference [9] presented the method for calculating the electric and magnetic fields from dipoles embedded in the formation, and [10] extended the method to calculate the electric and magnetic fields from arbitrarily oriented dipoles in the formation. Zhong used the coefficient propagator method to derive the magnetic field formula, assuming that the borehole effect and invasion zones can be neglected [11]. In forward modeling, it is very important to calculate the magnetic field formula at each measurement point. However, each of the magnetic field formulas contains the infinite integral of the Bessel function. Gao used the fast Hankel transform method to complete the computation of the indefinite integral of the Bessel function [12]. Zhang employed the Gauss quadrature algorithm to accelerate convergence speed [13]. It is easy to obtain non-convergent or even divergent results if the Bessel function is incorrectly calculated. Based on our earlier work [14], we combined the Gauss–Legendre quadrature with the continued fraction summation to solve the infinite integral of the Bessel function, and simulated the response of the LWD instrument with a different angle in the three-layer formation model. This method can cause the infinite integral quickly converge, and is well suited to calculate the infinite integral of the Bessel function in the magnetic field formulas.

Inversion modeling interprets the data measured by the LWD instrument as formation information and position information [15,16,17,18]. Various research has been carried out to solve the inversion problem. Yu developed a fast, robust and hands-off inversion method, and the method describes the layered formation using equal thickness layers [19]. Lu used the singular value decomposition method to perform an inversion algorithm [20]. Wang used the regularized Gauss–Newton method to acquire the distance to the formation boundary as well as the formation resistivity [4]. Heriyanto used the singular value decomposition and the Levenberg–Marquardt (LM) method to achieve a 1D resistivity inversion [21]. Pardo used the Gauss–Newton and many optimization methods to realize the 1D inversion of LWD [22]. In the process of LWD, it is very important not only to obtain the distance to the formation boundary, but also to judge whether the instrument is above or below the boundary in real time.

In inversion modeling, the LM algorithm combined with the line search method of the Armijo criterion is used to accurately achieve the inversion of the LWD tool in real time, and a constraint algorithm is added. In order to preserve the positive and negative sign of the distance parameters, the traditional logarithmic processing is replaced by the root of the fifth order in the data preprocessing of the initial parameters. We have done several sets of experiments to verify the convergence of the forward algorithm and the accuracy of the inversion. First, the convergence of the continued fraction summation and the direct summation in calculating magnetic field responses are compared. Then, the response of the LWD instrument in a layered formation is simulated, and its sensitivity to the formation boundary and the dipping angle is analyzed. Finally, the two-parameter and three-parameter inversion in two-layer formation is carried out.

## 2. Forward Modeling

### 2.1. Theory of Forward Modeling

On the basis of the traditional LWD instrument with axial coils, a transverse coil is added to the instrument to increase its azimuth detection characteristics. Figure 1 is a schematic diagram of the structure of the LWD logging tool used in this study. This apparatus consists of two axially symmetric compensation transmitting coils *Tr_1_* and *Tr_2_*, two axial receiving coils *Re_1_* and *Re_2_*, and a transverse receiving coil *Re_3_*. The axial coil and the transverse coil are perpendicular to each other. The axial transmitting coils and the axial receiving coils are symmetrical about the transverse coil. The symmetrical compensation coils of the instrument greatly reduce the effects of the formation environment during the measurement process [23]. They can also eliminate the influence of anisotropic formations and make the results more accurate. In Figure 1, O-XYZ is the formation coordinate system, O’-X’Y’Z’ is the instrument coordinate system, θ is the dipping angle, β is the azimuthal angle between the X-axis and the projection of the instrument on the X-Y plane, and γ is the rotation angle [24].

Coils can be represented by magnetic dipoles [25]. Therefore, the magnetic field responses emitted by the transmitting coils on each receiving coil can be obtained [26]. The electric fields and the magnetic fields from the unit magnetic dipole moment tensor ${\hat{M}}_{0}$ satisfy the Maxwell equation in this form [27], $\hat{H}$ is the magnetic field from the unit magnetic dipole in the formation coordinate system.
(1)$$\nabla \times \hat{E}=i\omega \mu_{0}\hat{H}+i\omega \mu_{0}{\hat{M}}_{0}$$
(2)$$\nabla \times \hat{H}=\hat{\sigma }\hat{E}+j\omega \hat{\varepsilon }\hat{E}$$

It is first necessary to calculate the magnetic field responses $\bar{H}$ in the formation coordinate system, and then convert it into the magnetic field responses $\bar{H}’$ in the instrument coordinate system. The relationship between the magnetic $\bar{M}$ of the formation coordinate system and the magnetic $\bar{M}’$ of the instrument coordinate system is shown in (3). The relationship between the magnetic field responses in two coordinate systems is shown in (5).
(3)$$\bar{M}=R\bar{M}’$$
(4)$$\bar{H}=\hat{H}\bar{M}$$
(5)$$\bar{H}’=R^{-1}\bar{H}=R^{-1}\hat{H}R\bar{M}’$$
(6)$$R=\left[\begin{array}{ccc} \cos\theta \cos\beta \cos\gamma -\sin\beta \sin\gamma & -\cos\theta \cos\beta \cos\gamma -\sin\beta \cos\gamma & \sin\theta \cos\beta \\ \cos\theta \sin\beta \cos\gamma +\cos\beta \sin\gamma & -\cos\theta \sin\beta \sin\gamma +\cos\beta \cos\gamma & \sin\theta \sin\beta \\ -\sin\theta \cos\gamma & \sin\theta \sin\gamma & \cos\theta \end{array}\right]$$

Several excellent reviews deriving these magnetic field formulas of the LWD tool in the layered formation are available and so these topics will not be discussed in detail here [13]. Each of the magnetic field formulas contains the infinite integral of the Bessel function [11,28]. It is easy to obtain non-convergent or even divergent results if the infinite integral is incorrectly calculated. The Gauss–Legendre quadrature combined with the continued fraction summation is used to solve this problem. Details of the Gauss–Legendre quadrature algorithm are presented in Appendix A [29,30]. After calculating the magnetic field on each receiving coil, we can obtain the phase difference between the axial coils *Re_1_* and *Re_2_*, and the voltage on the transverse coil *Re_3_*.
(7)$$V_{X}=i\omega \mu_{0}SH$$
(8)$$\Delta \varphi_{1}=\varphi_{1}-\varphi_{2}=\frac{180}{\pi }imag\left[\ln\left(\frac{V_{1}}{V_{2}}\right)\right]$$
(9)$$\Delta \varphi =\frac{\Delta \varphi_{1}+\Delta \varphi_{2}}{2}$$
where $V_{X}$ is the voltage on the receiving coil, S is the area of the receiving coils, $\Delta \varphi_{1}$ is the phase difference between *Re_1_* and *Re_2_* without compensation, and Δφ is the phase difference between *Re_1_* and *Re_2_* after compensation.

### 2.2. The Continued Fraction Summation

In the process of calculating the infinite integral of the Bessel function, if many single intervals are simply added up, the results may be non-convergent or even divergent. The continued fraction summation can very well solve this problem. The form of the continued fraction as follows:(10)$$S=\frac{d_{1}}{1+\frac{d_{2}}{1+\frac{\cdots }{1+\frac{d_{n-1}}{d_{n}}}}}$$

The coefficient $d_{i}$ can be calculated by a recursive algorithm [31]. Starting from the first coefficients,
(11)$$d_{1}=D_{1},\;\;\;\;\;D_{1}=p_{1}$$
(12)d2=−D2D1,     D1=p2
(13)d3=−D3D2,     D1=p3+p2d2
(14)d4=−D4D3,     D1=p4+p3(d2+d3)

When n≥5, the coefficients can be recursively solved:(15)$$L=2\cdot floor(\frac{n-1}{2})$$
(16)$$X(1)=d_{2},X(2)=d_{2}+d_{3}$$

Interchange X(1)↔X(2), and set X(L−1)=0, and we have
(17)$$X(k)=X(k-1)+d_{n-1}X(k-2),k=L,L-2,L-4,\ldots ,4$$

Then, $X(2)=X(1)+d_{n-1}$.

In the end, the coefficient $d_{i}$ can be obtained:(18)$$D_{n}=p_{n}+\sum_{i=1}^{L/2}p_{n-i}X(2i-1)$$
(19)dn=−DnDn−1

## 3. Inversion Modeling

### 3.1. Theory of Inversion Modeling

In inversion modeling, the data obtained from the LWD tool needs to be interpreted as the formation information [32]. It is very important to accurately obtain the position of the instrument and the formation resistivity in real time. In this study, a positive and negative sign is added to the distance parameter to distinguish whether the instrument is above or below the formation boundary. Preprocessing the unknown parameters can improve the efficiency of the inversion. In order to retain the sign of inversion parameters, we replaced the traditional logarithmic processing with the 5th root in the data preprocessing of the unknown parameters.
(20)$$M={\left[m_{1},m_{2},\cdots ,m_{N}\right]}^{T}$$
(21)$$x={\left[\sqrt[{5}]{x_{1}},\sqrt[{5}]{x_{2}},\cdots ,\sqrt[{5}]{x_{n}}\right]}^{T}$$
where the vector M represents the measured data, and the vector x represent the unknown parameters [13]. N represents the amount of data measured at each measurement point, and n represents the number of unknown parameters that need to be inverted.

The main purpose of the inversion is to minimize the cost function C(x), which has the form
(22)$$C\left(x\right)=\frac{1}{2}R{\left(x\right)}^{T}R\left(x\right)$$
where R(x)=F(x)−M, and F(x) is the vector of simulated tool response values generated by forward modeling.

In order to minimize the cost function C(x), the LM algorithm is used [33]. The Taylor series expansion is expressed for cost function and approximated by the Jacobian matrix J as follows [34]

(23)$$C(x)\approx \frac{1}{2}R^{T}\left(x_{c}\right)R\left(x_{c}\right)+R^{T}\left(x_{c}\right)J\left(x_{c}\right)\left(x-x_{c}\right)+\frac{1}{2}{\left(x-x_{c}\right)}^{T}\left(J^{T}\left(x_{c}\right)J\left(x_{c}\right)+\mu I\right)\left(x-x_{c}\right)$$

The solution of the equation is
(24)$$x_{k+1}=x_{k}-{\left(J^{T}\left(x_{k}\right)J\left(x_{k}\right)+\mu I\right)}^{-1}J^{T}\left(x_{k}\right)R\left(x_{k}\right)$$
(25)$$d_{k}=x_{k+1}-x_{k}=-{\left(J^{T}\left(x_{k}\right)J\left(x_{k}\right)+\mu I\right)}^{-1}J^{T}\left(x_{k}\right)R\left(x_{k}\right)$$
where J is the M×N Jacobian matrix consisting of the first-order derivative of the data mismatch, and $J_{ij}=\partial F_{i}/\partial x_{j}$ is approximated with the finite difference in the inversion. 

Parameter μ is adjusted with a strategy similar to adjusting the radius of the trust region, that is, the LM algorithm [33,35]. The detailed method of obtaining the search direction $d_{k}$ by updating the parameter μ is as follows. We define a quadratic function at the iteration point [36].
(26)$$Q\left(d\right)=F\left(x_{k}\right)+d_{k}^{T}\nabla F\left(x_{k}\right)+\frac{1}{2}d_{k}^{T}J_{k}^{T}J_{k}d_{k}$$

We use $\xi_{k}$ to represent the ratio of the two increments:(27)$$\xi_{k}=\frac{\Delta F\left(d_{k}\right)}{\Delta Q\left(d_{k}\right)}=\frac{F\left(x_{k+1}\right)-F\left(x_{k}\right)}{d_{k}^{T}\nabla F\left(x_{k}\right)+\frac{1}{2}d_{k}^{T}J_{k}^{T}J_{k}d_{k}}$$

We give μ an initial value, and then continuously adjust the μ according to the value of $\xi_{k}$ to calculate $d_{k}$. The update rules are shown in Table 1.

Usually, the step $d_{k}$ cannot give the minimum value of the cost function C(x). To overcome this problem, a line search approach is used to find a step λ along $d_{k}$. Using the line search method of the Armijo criterion to update the search step, $m_{k}$ can be reached as an integer value satisfying an equation as follows:(28)$$C\left(x_{k}+\lambda d_{k}\right)\leq C\left(x_{k}\right)+\sigma \rho^{m_{k}}g_{k}^{T}d_{k}$$
where $\lambda =\rho^{m_{k}}$, $g_{k}=J^{T}\left(x_{k}\right)R\left(x_{k}\right)$, and $\sigma =10^{-4}$. 

### 3.2. The Constraint Algorithm

In order to avoid the abnormal inversion results and improve the stability of the inversion, a constraint algorithm is added. The intermediate parameter c is added instead of the parameter x.
(29)$$\begin{array}{cc} x=x_{\max}-\frac{x_{\max}-x_{\min}}{c^{2}+1}, & -\infty <c<+\infty \end{array}$$
(30)$$\begin{array}{cc} x\to x_{\max}, & \begin{array}{cc} as & c\to \pm \infty \end{array}\\ x\to x_{\min}, & \begin{array}{cc} as & c\to 0 \end{array} \end{array}$$
where $x_{\max},x_{\min}$ are the upper and lower bounds of the parameter x, respectively. In the Jacobian matrix, it can be expressed as follows:(31)$$\frac{\partial F}{\partial c}=\frac{\partial F}{\partial x}\frac{dx}{dc}=2\frac{\partial F}{\partial x}\frac{x-x_{\max}}{x_{\min}-x_{\max}}\sqrt{(x-x_{\max})\left(x_{\min}-x\right)}$$

The two successive iterates $x_{k+1}$ and $x_{k}$ of x are related by
(32)$$x_{k+1}=x_{\max}-\frac{x_{\max}-x_{\min}}{c_{k+1}^{2}+1}=x_{\max}-\frac{x_{\max}-x_{\min}}{{\left(c_{k}+q_{k}\right)}^{2}+1}$$
where $c_{k}=\sqrt{\frac{x_{k}-x_{\min}}{x_{\max}-x_{k}}}$, and $q_{k}=c_{k+1}-c_{k}$ is the search step in c towards the minimum of the cost function C(x). The relationship between $d_{k}$ and $q_{k}$ is as follows:(33)$$d_{k}=q_{k}\frac{dx_{k}}{dc_{k}}$$

In order not to specifically calculated the parameters c and q, the round-off errors caused by the introduction of the nonlinear function are reduced, the relationship between the two successive iterates $x_{k+1}$ and $x_{k}$ of x can be obtained:(34)$$x_{k+1}=x_{\min}+\frac{x_{\max}-x_{\min}}{\alpha_{k}^{2}+\left(x_{k}-x_{\min}\right){\left(x_{\max}-x_{k}\right)}^{3}}\alpha_{k}^{2}$$
where
(35)$$\alpha_{k}=\left(x_{k}-x_{\min}\right)\left(x_{\max}-x_{k}\right)+\frac{1}{2}\left(x_{\max}-x_{\min}\right)\lambda d_{k}$$

## 4. Results and Discussions

In this section, the convergence of the continued fraction summation and the direct summation in calculating magnetic field formulas are compared. Then, the response of the LWD instrument in the layered formation model is simulated, and the sensitivity of the LWD instrument in the formation boundary and different dipping angle is analyzed. Finally, two-parameter and three-parameter inversions in the two-layer formation model are carried out.

### 4.1. Convergence Comparison between Continued Fraction Summation and Direct Summation

It is necessary to accumulate its integral intervals when calculating the infinite integral of the Bessel function in magnetic field formulas. The summation of integral intervals is calculated by direct summation and continued fraction summation. In order to see the convergence results in the calculation process, we compared the convergence of the infinite integral of the Bessel function in magnetic field formulas of the axial receiving coil with that of the transverse receiving coil by the continued fraction summation and the direct summation, respectively. Four observation points were selected in the formation to observe the change of the results of the magnetic field responses with the number of integral intervals. The results on the axial receiving coil and the transverse receiving coil are shown in Figure 2 and Figure 3, respectively. In Figure 2 and Figure 3, the horizontal axis represents the number of integral intervals and logging sample points, and the vertical axis represents the magnitude of the magnetic field. The red lines represent the results of the continued fraction summation, and the blue dotted lines represent the results of the direct summation.

It can be seen from Figure 2 and Figure 3 that, for the red lines the magnetic field responses remain unchanged after three or four integral intervals and are accumulated. For the blue dotted lines, the magnetic field responses remain unchanged after about six integral intervals and are accumulated. It can be concluded that the continued fraction summation converges faster than the direct summation when calculating the magnetic field formulas.

### 4.2. The Simulation of Forward Modeling

After that, we simulated the response of the LWD instrument with different dipping angles in the three-layer formation model. In this formation model, the resistivity of the first layer was 2 Ω⋅m and the resistivity of the second layer was 10 Ω⋅m, while the resistivity of the third layer was 2 Ω⋅m, and the thickness of the intermediate layer was 3.6 m. The frequencies used were 400 kHz and 2 MHz. According to the theory of forward modeling, the magnetic field of the transmitting coil at the receiving coil in the formation coordinate system can be calculated. The Gauss–Legendre quadrature was used to calculate the Bessel function in the magnetic field equation. After that, the magnetic field in the formation coordinate system was transformed into the magnetic field in the instrument coordinate system according to the angle of the instrument. In this way, the phase difference between the two axial receiving coils and the voltage on the transverse receiving coil were calculated. The simulation was performed in MATLAB software. The simulated results after compensation are shown in Figure 4 and Figure 5. In Figure 4 and Figure 5, the horizontal axis represents the distance between the instrument and the upper boundary; a positive sign indicates that the instrument is below the formation boundary, and a negative sign indicates that the instrument is above the formation boundary. The vertical axis in Figure 4 represents the magnitude of the phase difference between the axial receiving coils. The vertical axis in Figure 5 represents the magnitude of the voltage of the transverse receiving coil.

Figure 4a,b shows the phase difference between the axial receiving coils at a frequency of 400 KHz and 2 MHz. It can be seen that there will be mutations at the formation boundary, and as the formation resistivity changes, the magnitude of the phase difference also changes. Figure 5a,b shows the voltage of the transverse receiving coil at a frequency of 400 KHz and 2 MHz. It can be seen that the closer the LWD instrument is to the formation boundary, the larger the voltage of the transverse receiving coil. The farther the LWD instrument is from the formation boundary, the closer the voltage of the transverse receiving coil is to zero. The azimuth response characteristics in different directions will be produced when the instrument enters the low resistivity layer from the high resistivity layer or the instrument enters the high resistivity layer from the low resistivity layer. Therefore, it can distinguish whether the LWD instrument is entering the layer or going out of the layer.

In addition, it can be seen from Figure 4 and Figure 5 that with an increase of the dipping angle, whether with a phase difference response or a voltage response, the peaks on the formation boundary become more and more obvious. The larger the dipping angle, the more sensitive the LWD instrument is to the formation boundary. 

### 4.3. Results of Inversion Modeling

Finally, we performed two-parameter and three-parameter inversions in the two-layer formation model. The two-layer formation model is shown in Figure 6. Here, a positive and negative sign is added to the unknown parameter *d*, the distance from the formation boundary. A positive sign indicates the instrument is below the formation boundary and a negative sign indicates the instrument is above the boundary.

First, we performed a two-parameter inversion in the two-layer formation model. The phase difference between axial receiving coils *Re_1_* and *Re_2_* at a frequency of 400 kHz and 2 MHz, and the voltage of the transverse coil *Re_3_* at a frequency of 400 kHz, were used as measured data. The parameters to be inverted were the resistivity of the lower layer *R2* and the distance *d* of the instrument from the formation boundary. *R1* was 2 Ω⋅m. The dipping angle was 80°. The measured data vector M and the unknown parameters x are shown below. We use the 5th root in the data preprocessing of the unknown parameters x.
(36)$$M={\left[\Delta \varphi_{400kHz},\Delta \varphi_{2MHz},V_{X}\right]}^{T}$$
(37)$$x={\left[\sqrt[{5}]{R_{2}},\sqrt[{5}]{d}\right]}^{T}$$

Seven formation samples were randomly selected to test the accuracy of the inversion. We set the initial values for the inversion and calculate the cost function. Accurate inversion results can be obtained by continuously reducing cost function. The true values, initial values, iterations and inversion results in the two-parameter inversion are shown in Table 2. When setting the initial values, many situations were considered, such as whether the initial value of the boundary distance parameter is the same or opposite to the positive and negative sign of the real value, and whether the initial resistivity parameter is larger or smaller than the real value. The cost function of inversion samples varying with the number of iterations in the two-parameter inversion are shown in Figure 7. 

Finally, the three-parameter inversion was performed in the two-layer formation model. The phase difference between *Re_1_* and *Re_2_* at a frequency of 400 kHz and 2 MHz, and the voltage of the transverse coil *Re_3_* at a frequency of 400 kHz, were used as measured data. The parameters to be inverted were the resistivity of the upper layer *R1* and lower layer *R2* and the distance *d* of the instrument from the boundary. The angle of the instrument was 80°. The measured data vector M and the unknown parameters x are shown below. We used the 5th root in the data preprocessing of the unknown parameters x.
(38)$$M={\left[\Delta \varphi_{400kHz},\Delta \varphi_{2MHz},V_{X}\right]}^{T}$$
(39)$$x={\left[\sqrt[{5}]{R1},\sqrt[{5}]{R2},\sqrt[{5}]{d}\right]}^{T}$$

Seven formation samples were randomly selected to test the accuracy of the inversion. We set the initial values for the inversion and calculate the cost function. Accurate inversion results can be obtained by continuously reducing cost function. The true values, initial values, iterations and inversion results in the three-parameter inversion are shown in Table 3. When setting the initial values, many situations were considered, such as whether the initial value of the boundary distance parameter is the same or opposite to the positive and negative sign of the real value, and whether the initial resistivity parameter was larger or smaller than the real value. The cost function of inversion samples varying with the number of iterations in the three-parameter inversion are shown in Figure 8.

From Table 2 and Table 3, it can be seen that when the initial values are set close to the real values, the accurate results can be quickly obtained. When the initial values are far from the real value, accurate results can be obtained through multiple iterations. In each sample, accurate inversion results can be obtained without abnormal value. As can be seen from Figure 7 and Figure 8, the cost function of all samples rapidly decreased with the number of iterations. This showed that the inversion can continuously optimize the results and finally obtain accurate inversion results. When predicting the boundary distance, it can not only accurately predict its value, but also judge whether it is above or below the boundary. It can accurately achieve geological guidance.

## 5. Conclusions

In this study, based on fast-forward modeling, an efficient inversion, which can accurately and stably invert the information of the layered formation in real time, was described. In forward modeling, the magnetic field responses were quickly and accurately calculated by the Gauss–Legendre quadrature method and continued fraction summation. The summation of continued fraction summation can converge faster than that of direct summation in calculating the infinite integral of the Bessel function in magnetic field formulas. In the inversion, the LM algorithm combined with the line search method of the Armijo criterion was used, and a convergence algorithm was added to it. The two-parameter and three-parameter inversion of the LWD instrument in the two-layer model was performed. The inversion not only obtained the distance from the formation boundary and the formation resistivity in real time, but it also accurately judged whether the LWD instrument was above or below the boundary. The results of the experiment indicate the inversion can accurately and stably obtain the formation information. It has great significance for LWD in geo-steering.

## Figures and Tables

**Figure 1 sensors-19-02754-f001:**
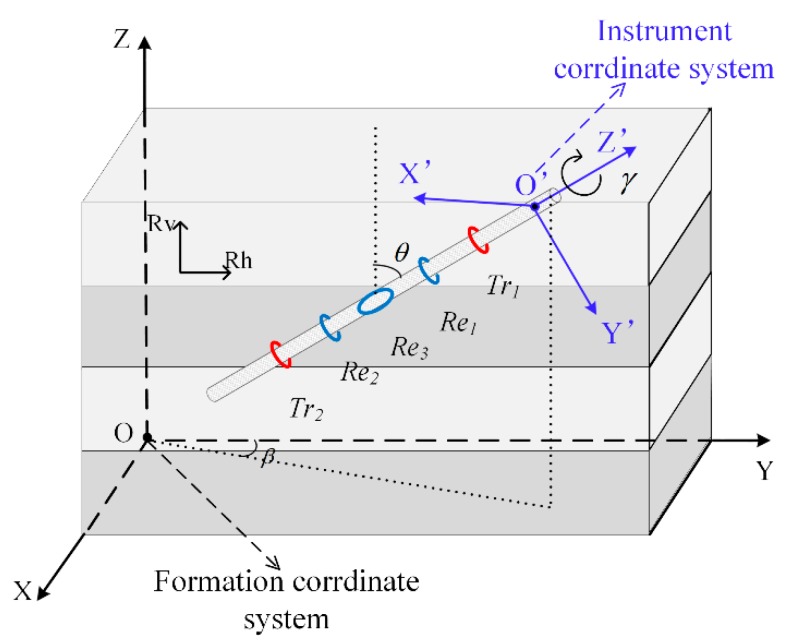
Schematic diagram of the logging while drilling (LWD) instrument used in this study.

**Figure 2 sensors-19-02754-f002:**
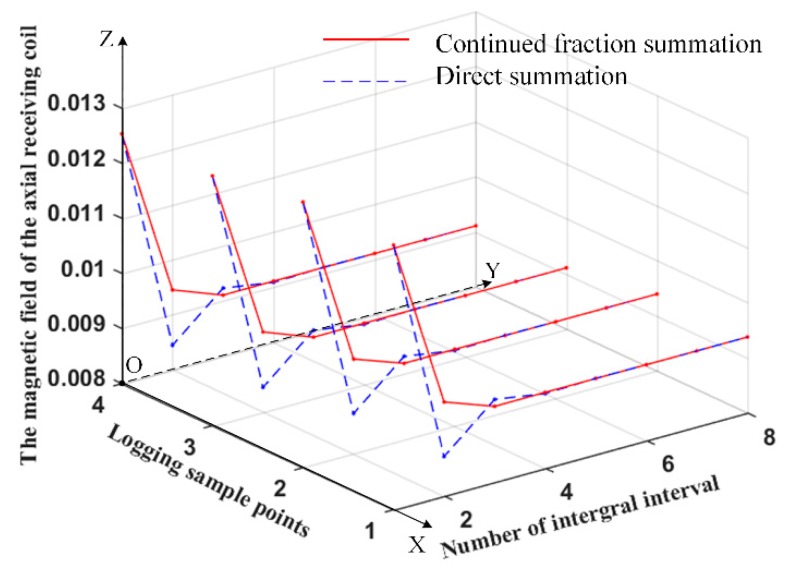
The results of the magnetic field responses on the axial receiving coil with the number of integral intervals.

**Figure 3 sensors-19-02754-f003:**
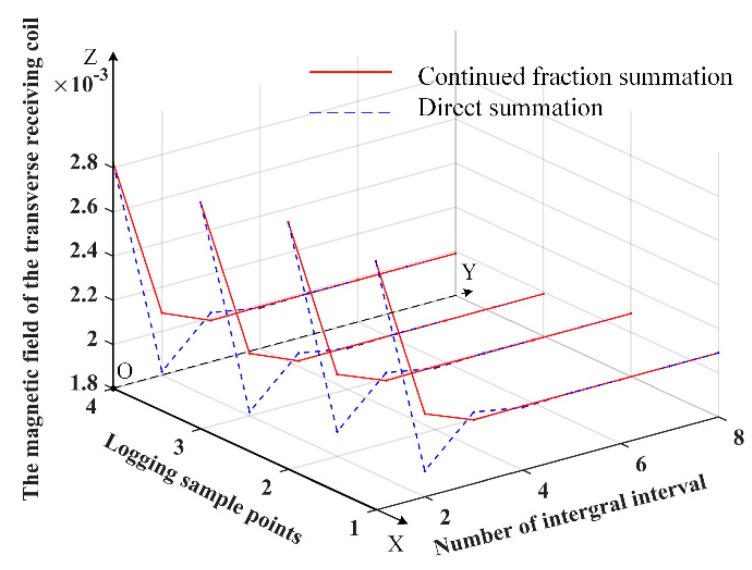
The results of the magnetic field responses on the transverse receiving coil with the number of integral intervals.

**Figure 4 sensors-19-02754-f004:**
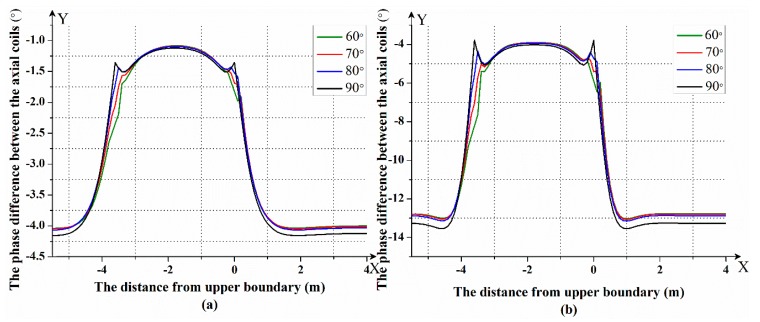
The phase difference between the axial receiving coils at a frequency of (**a**) 400 kHz and (**b**) 2 MHz.

**Figure 5 sensors-19-02754-f005:**
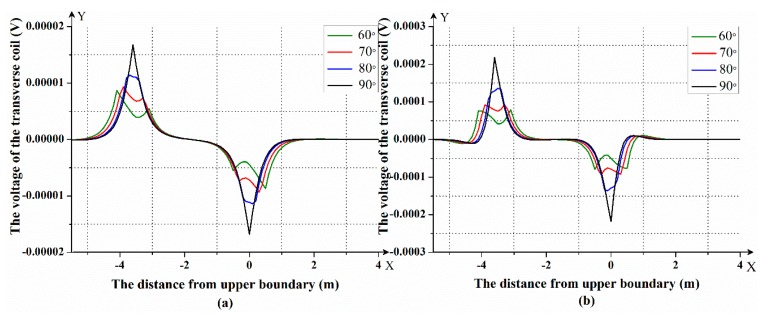
The voltage of the transverse receiving coil *Re_3_* at a frequency of (**a**) 400 KHz and (**b**) 2 MHz.

**Figure 6 sensors-19-02754-f006:**
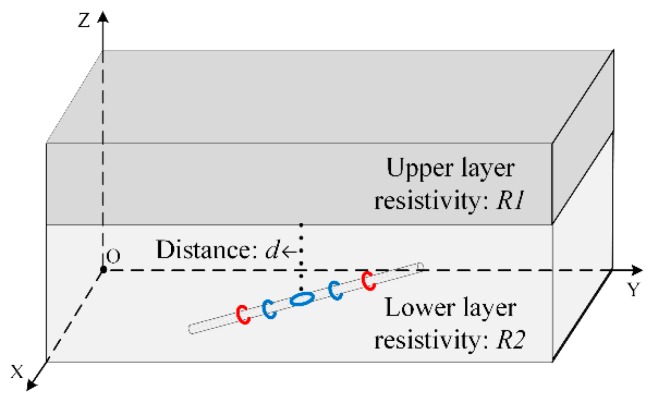
Two-layer formation model diagram.

**Figure 7 sensors-19-02754-f007:**
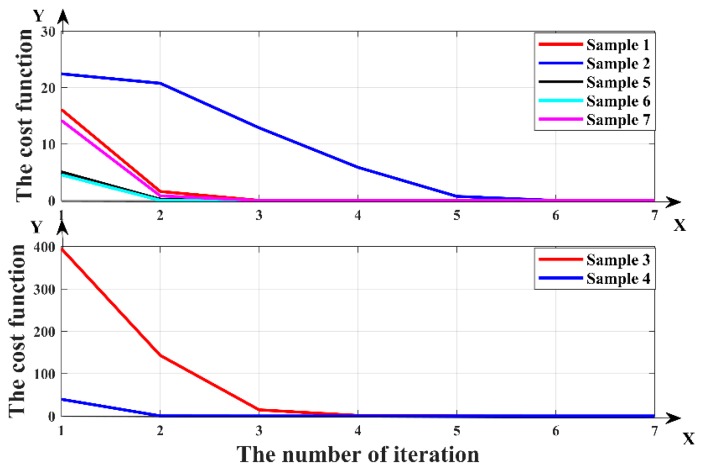
The cost function of the two-parameter inversion varying with the number of iterations.

**Figure 8 sensors-19-02754-f008:**
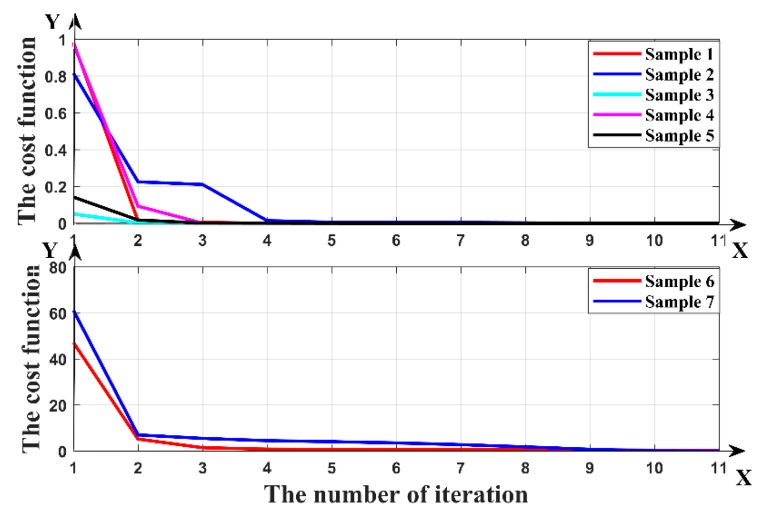
The cost function of the three-parameter inversion varying with the number of iterations.

**Table 1 sensors-19-02754-t001:** Update rule for parameter μ.

$\xi_{k}\geq 0.75$	$\mu_{k+1}=0.1\mu_{k}$
$0.25<\xi_{k}<0.75$	$\mu_{k+1}=\mu_{k}$
$\xi_{k}\leq 0.25$	$\mu_{k+1}=10\mu_{k}$

**Table 2 sensors-19-02754-t002:** Results of two-parameter inversion in the two-layer formation model.

Sample	True Values		Initial Values		Iterations	Inversion Vesults	
	*R2* (Ω∙m)	*d* (m)	*R2* (Ω∙m)	*d* (m)		*R2* (Ω∙m)	*d* (m)
1	10.00	−0.40	4.00	−0.30	5	10.00	−0.40
2	10.00	−0.40	4.00	−0.10	7	10.00	−0.40
3	10.00	0.10	5.00	0.40	6	10.00	0.10
4	10.00	−0.10	6.00	0.70	5	10.00	−0.10
5	4.00	0.20	6.00	0.50	5	4.00	0.20
6	8.00	−0.20	10.00	−0.50	4	8.00	−0.20
7	8.00	−0.10	10.00	−0.70	5	8.00	−0.10

**Table 3 sensors-19-02754-t003:** Results of three-parameter inversion in the two-layer formation model.

Sample	True Values			Initial Values			Iterations	Inversion Results		
	*R1* (Ω∙m)	*R2* (Ω∙m)	*d* (m)	*R1* (Ω∙m)	*R2* (Ω∙m)	*d* (m)		*R1* (Ω∙m)	*R2* (Ω∙m)	*d* (m)
1	5.00	18.00	0.20	10.00	13.00	0.40	5	5.00	18.00	0.20
2	5.00	18.00	0.20	8.00	13.00	−0.40	11	5.00	18.00	0.20
3	10.00	18.00	−0.20	8.00	13.00	−0.40	5	10.00	18.00	−0.20
4	10.00	15.00	−0.10	13.00	19.00	−0.30	4	10.00	15.00	−0.09
5	10.00	15.00	0.10	15.00	17.00	0.20	5	10.00	15.00	0.10
6	10.00	5.00	−0.10	15.00	17.00	0.03	15	10.00	5.00	−0.10
7	11.00	3.00	−0.50	13.00	19.00	0.00	11	10.99	3.00	−0.50

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
