# Peer review of "Application of Logging While Drilling Tool in Formation Boundary Detection and Geo-steering"

_sensors, 2019, doi:10.3390/s19122754_

Round 1

Reviewer 1 Report

Logging while drilling (LWD) is an important realtime geo-steering and monitoring tool based on magnetic or acoustic image synthesis techniques. Fast forward modeling and data processing plays an important role in Oil Geophysical Prospecting these days. The reviewer is very glad to see the progresses about inversion modeling and related applications by the author's group. The manuscript is well organized and presented.  The following are comments and critique by the reviewer:

General and Major Comments:

1, the manuscript discussed several topics like forward modeling and inversion modeling, there are some previous papers introduced these topics and related algorithms, please add some general review information about progresses of forward modeling and inversion modeling algorithms besides how these methods are used in LWD, please also add some background information for optimization methods if possible.

2, some calculation methods (E.g., integral value calculation and simplified Bessel function in 2.2) has already been introduced in the authors’ other papers (Also other groups’ papers). Please consider reducing the paragraph and including the calculations in the appendix part.

3, please add more details for 4.2 (simulation of forward modeling). E.g., simulation platform, a short review of simulation work has been done by other groups, appendix for parameters and techniques used in your simulation, etc. You can also add a picture to demonstrate how you performed the simulation.

4, the review would recommend revising your figure 6 and put some double arrow signs to indicate where are your R1, R2 and d preciously. Please also add xyz coordinate system the same as your figure 1 to get all the figures consistent with each other.

Minor Comments:

1, please pay attention to spelling and grammar mistakes. E.g., “Application of the Logging While Drilling Tool in Formation Boundary Detection and Geo-steering”, “Second After that, we simulated the response of the LWD instrument with different dipping angles in the three-layer formation model.”

2, please change some sentences which are not described accurately. E.g., “The coils can be replaced represented by magnetic dipoles”  

Author Response

Response to Reviewer 1 Comments

Point 1: Logging while drilling (LWD) is an important real-time geo-steering and monitoring tool based on magnetic or acoustic image synthesis techniques. Fast forward modeling and data processing plays an important role in Oil Geophysical Prospecting these days. The reviewer is very glad to see the progresses about inversion modeling and related applications by the author's group. The manuscript is well organized and presented.  The following are comments and critique by the reviewer:

Response 1: We greatly appreciate your encouraging comments and thank you for your insightful suggestions. We are sorry for the missing background information on this part. We believe that the review information mentioned by reviewers can help readers understand the progress of forward modeling and inversion modeling more easily.

Author action:According to your comments, we updated the manuscript by adding some review information about the progress of forward modeling and inversion modeling.

Point 2: some calculation methods (E.g., integral value calculation and simplified Bessel function in 2.2) has already been introduced in the authors’ other papers (Also other groups’ papers). Please consider reducing the paragraph and including the calculations in the appendix part.Response 2: Thank you for your good advice. We believe that the reviewer’s suggestion is beneficial to the article.

Author action: According to your comments, we updated the manuscript by incorporating some published calculation methods into the Appendix.

Point 3: please add more details for 4.2 (simulation of forward modeling). E.g., simulation platform, a short review of simulation work has been done by other groups, appendix for parameters and techniques used in your simulation, etc. You can also add a picture to demonstrate how you performed the simulation.Response 4: Thank you for the pertinent and professional query on this issue. We are sorry for missing an explanation about this issue. The forward simulation is detailed as follows. According to the forward modeling theory, the magnetic field of the transmitting coil at the receiving coil in the formation coordinate system can be calculated from the known formation information. The Gauss-Legendre quadrature method is used to calculate the Bessel function in the magnetic field equation. After that, the magnetic field in the formation coordinate system is transformed into the magnetic field in the instrument coordinate system according to the angle of the instrument. By this way, the phase difference between the two axial receiving coils and the voltage on the transverse receiving coil are calculated. The simulation is performed in MATLAB software.

Author action: According to your comments, we updated the manuscript by adding details of forward simulation in Section 4.2.

Point 4: the review would recommend revising your figure 6 and put some double arrow signs to indicate where are your R1, R2 and d preciously. Please also add xyz coordinate system the same as your figure 1 to get all the figures consistent with each otherResponse 4: Thank you for your advice. We are sorry for lacking the demonstration of the parameter information clearly in Figure 6 and sorry for not adding the XYZ coordinate system to other figures except for Figure 1. In Figure 6, R1 represents the upper resistivity, R2 represents the lower resistivity, and d represents the distance from the instrument to the interface.

Author action: According to your comments, we updated the manuscript by demonstrating the parameter information in Figure 6 and adding the XYZ coordinate system to all the pictures in this paper.

Point 5: Minor Comments:1, please pay attention to spelling and grammar mistakes. E.g., “Application of the Logging While Drilling Tool in Formation Boundary Detection and Geo-steering”, “Second After that, we simulated the response of the LWD instrument with different dipping angles in the three-layer formation model.”2, please change some sentences which are not described accurately. E.g., “The coils can be replaced represented by magnetic dipoles”Response 5: Thanks a lot for your scrupulous review of our manuscript. We are sorry for our mistakes. We have gone through our manuscript carefully, modified spelling and grammar mistakes and changed some inaccurate sentences.

Author action: According to your comments, we have gone through our manuscript carefully, modified spelling and grammar mistakes and changed some inaccurate sentences.

Reviewer 2 Report

An efficient inversion modeling for LWD proposed by the author was discussed in this paper, and the validity of proposed method was investigated. This paper will be very useful for the readers in the related field.

However, this paper seems to lack some information and the reviewer requires the minor revision to improve the quality of this paper.

The detailed comments are listed as follows.

1. In Table 2, when the true values of R2 and d are10.00 and 0.10, the inversion results were 10.00 and -1.00. Why was the d of inversion value far different from that of the true value. This phenomenon was not observed in the other inversion results shown in Tables 2 and 3 and there is no discussion about this. The reviewer recommends to add the discussion about this disagreement.

2. The legend should be added to Figs. 7 and 8 because there is no statement what the color lines show.

3. The author uses R1 and R2 for both the coil name and the resistivity of layer. This is confusing and please change one parameter to the different characters.

Author Response

Response to Reviewer 2 Comments 

Point 1: An efficient inversion modeling for LWD proposed by the author was discussed in this paper, and the validity of proposed method was investigated. This paper will be very useful for the readers in the related field. However, this paper seems to lack some information and the reviewer requires the minor revision to improve the quality of this paper. The detailed comments are listed as follows. 1. In Table 2, when the true values of R2 and d are10.00 and 0.10, the inversion results were 10.00 and -1.00. Why was the d of inversion value far different from that of the true value. This phenomenon was not observed in the other inversion results shown in Tables 2 and 3 and there is no discussion about this. The reviewer recommends to add the discussion about this disagreement. 

Response 1: We greatly appreciate your encouraging comments and thank you for your insightful suggestions. We are very sorry for our incorrect writing “0.10” as "-1.00". In order to ensure the validity and rigor of the result, we have re-simulated the case and the result is “10, 0.10”. 

Author action: According to your comments, we updated the manuscript by correcting Table 2. 

Point 2: The legend should be added to Figs. 7 and 8 because there is no statement what the color lines show. 

Response 2: Thank you for the pertinent and professional query on this issue. We are very sorry for not adding legends in Figs. 7 and 8. A line in Figure 7 illustrates how the cost function of a sample in Table 2 varies with the iteration number in the inversion process increased. A line in Figure 8 illustrates how the cost function of a sample in Table 3 varies with the iteration number in the inversion process increased. In order to distinguish each sample, different line colors are set. 

Author action: According to your comments, we updated the manuscript by adding legends in Figures 7 and 8. 

Point 3: The author uses R1 and R2 for both the coil name and the resistivity of layer. This is confusing and please change one parameter to the different characters. 

Response 3: Thanks a lot for your scrupulous review of our manuscript. We are sorry for lacking a clear distinction between the coil name and the resistivity. In order to distinguish these two names more clearly, we use ‘Re1’, ‘Re2’, ‘Re3’, ‘Tr1’, ‘Tr2’ to represent the coil name, and ‘R1’, ‘R2’ to represent the resistivity. Thank you again for your suggestions. 

Author action: According to your comments, we updated the manuscript by using the coil name as ‘Re1’, ‘Re2’, ‘Re3’, ‘Tr1’, ‘Tr2’, and the resistivity as ‘R1’, ‘R2’.
